# Analyzing the Relationship between the Features of Direct Real Estate Assets and Their Corresponding Australian—REITs

Xinyi Li [1], Yuhong Zhang [2], Xing Zhang [3],* and Runtang Gu [3]

1   School of Architecture and Built Environment, The University of Adelaide, Adelaide, SA 5000, Australia
2   Faculty of Business, City University of Macau, Macau 999078, China
3   School of Management, Guizhou University of Commerce, Guiyang 550001, China
*   Correspondence: b19092100063@cityu.mo

**Abstract:** This study investigated the relationship between a sector-specific Australian Real Estate Investment Trust (A-REITs) and the underlying property assets in its property portfolio. The existing studies have assessed the connectedness/correlation between the A-REITs market and a variety of other asset markets, including the overall stock, bond, and direct real estate markets. This study applied regression analysis methods and discovered that there exists a certain degree of linear correlation between the underlying property assets and the return of the subject A-REITs. The most significant variable is the occupancy of the offices. The higher the occupancy is, the better the dividend can be. Features of the A-REITs also affect the dividend outcomes, specifically, the total portfolio market value and the capitalization rate. This suggests that the annual valuation outcomes show a positive relation with the performance of the A-REITs.

**Keywords:** REITs; property assets; property portfolio; real estate market

## 1. Introduction

Among investment vehicles such as equities, bonds, and foreign exchange transactions, real estate assets have been regarded as excellent diversification vehicles for portfolios. Individual investors are unlikely to make direct investments in commercial real estate assets such as office buildings, shopping complexes, and logistic distribution hubs. First, the amount of cash necessary to invest in these types of real estate assets is often substantial. Second, effective real estate asset investment needs both financial/economic expertise and specialized understanding of fields such as property development, construction management, and property asset management. In addition, transactions involving such real estate assets are frequently conducted off-market under normal conditions; therefore, the information efficiency within this particular industry cannot be guaranteed.

Real Estate Investment Trusts (REITs), formerly known as Listed Property Trusts (LPT) in Australia, provide an alternative to direct real estate investment, which requires participants to have a sufficient amount of capital, thereby lowering the entry barrier to the real estate market for individual investors. To provide dividends for the trust's shareholders, REITs rely on reliable and ongoing rental income. The unit price of REITs is influenced by both the underlying property and the stock market. In light of these characteristics, REITs can provide more asset liquidity than direct real estate investing. Meanwhile, their business strategy produces a risk-adjusted investment vehicle comparable to conventional stock/security investments.

Studies (Yong and Pham 2015; Tsai et al. 2010; West and Worthington 2006) of the relationship amongst REITs, stock, and direct real estate have confirmed that the REITs market is highly correlated to the stock market in the short term, normally within three years. As for the long-term variation trend, REITs shows the consistency of direct real estate. However, the leading studies are confined to exploring the phenomenon at the macro market level, with the application of large-scale market indices data.

### 1.1. Aims of Study

As a derivative of direct property investment, it has been shown that the REITs market has a long-term parallel relationship with the direct property investment industry. REITs are real estate corporations, regardless of their hybrid nature, since their tax-exempt status restricts them to real estate-related assets and operations. As REITs, particularly big REITs, are vertically integrated real estate enterprises, the foundation of REITs' business is property-related operations (Geltner et al. 2014).

This study aims to examine the relationship between the performance of the underlying assets of an individual A-REIT and the performance of other A-REITs in order to gain a deeper understanding of this connection. This study differs from the previous research in that it focuses on the detailed-level determinants of an individual A-REIT's underlying assets. This study has chosen the Centuria Office REIT (ASX Code: COF) as the object of study.

### 1.2. Research Questions

Based on the income generation of an A-REIT, the primary question of this study is: What is the correlation between the market performance of the underlying assets and their corresponding REIT? To answer the primary question, the following questions will be considered:

RQ1: Does the price/rent rate of the underlying assets also have a relationship with the unit price of the corresponding A-REIT?

RQ2: What is the relationship between the capitalization rate of the underlying assets and the corresponding A-REIT?

RQ3: Does the return rate of the underlying assets share the same variation tendency with the return of the corresponding A-REIT?

RQ4: If the abovementioned two indices share the same variation trend but at different scales, what is the cause?

## 2. Literature Review

### 2.1. Australian Real Estate Investment Trust

Australian Real Estate Investment Trusts (A-REITs), previously known as Australian Listed Property Trusts (LPTs), were established in 1971. The A-REITs market is now a well-established, comprehensive REITs market with 49 listed REITs on the Australian Stock Exchange (ASX) accounting for a market capitalization of $179.1 billion (ASX n.d.). The S&P/ASX 200 is the major index that tracks the A-REITs' market movement.

A-REITs have grown quickly in the last two decades, in spite of the adverse impact from the Global Financial Crisis in 2007–2008, which decreased the A-REITs market by 50% in market capitalization. As a result of the inflation-hedging feature of real estate, A-REITs recovered and reached a new peak in 2012. At that time, Australia was positioned as the world's second largest REITs market (by market capitalization) (Newell 2013). Following the shock of COVID-19 during 2020–2021, A-REITs' average dividend yield has remained at 3.4% since December 2021, double the 10-year government bond rates of Australia (Zhang 2022).

### 2.2. Studies on A-REITs

The substantial market volume of REITs and A-REITs, along with their characteristics as derivatives of direct real estate investment, have piqued the interest of academics and sparked extensive discussion on several REIT-related topics. According to the existing literature, REIT-related research has concentrated on four primary topics:

- The relationship between REITs and other traditional investment assets classes, e.g., direct real estate investment, securities, and bonds (Danso 2022; Nguyen et al. 2022; Hoesli and Oikarinen 2012; Giannarelli and Tiwari 2021; Boudry et al. 2012; Giliberto 1990).

- The evaluation of REITs' performance indices, or commentary on REITs' roles as substitutive investment vehicles, focusing on the prominent features including risk-adjustment, portfolio diversification, and market sensitiveness (Soyeh et al. 2021; Zhu et al. 2020; Eibel 2020; Sari et al. 2020; Lin et al. 2019; Erol and Tyvimaa 2019; Glabadanidis 2014; Lee and Lee and Lee 2012; Chikolwa 2011).
- The administration and management of REITs (Mintah et al. 2020; Parker 2016; Lee 2009).
- The contribution of REITs in achieving Environmental, Sustainability and Governance (ESG) goals (Vieira de Castro et al. 2020; Siew 2015).
- The first three categories have been thoroughly investigated, especially with the assistance of econometric methods. The last one is relatively more novel and is less related to this study. The detailed illustration of the existing studies is presented in the subsequent Literature Review Section.

### 2.3. Two Important Time Periods

The first Australian Real Estate Investment Trust (A-REITs) was listed on the Australian Stock Exchange (ASX) in 1971. As of the end of December 2021, A-REITs had reached $179.10 billion in market capitalization, the largest being ASX investment fund segments, followed by exchange-traded products (ETPs), and infrastructure. The market cap of A-REITs accounts for approximately 39% of the total ASX funds market but only accounts for 7% of the number of listed trusts. The average daily volume and transactions are both at the top amongst all ASX fund asset classes (ASX 2021).

Supported by such large trading volumes and active transactions, A-REITs-related studies have been well-established by academia. Before the outbreak of the COVID-19 pandemic at the commencement of 2020, there were two critical time periods that concerned researchers: The first was 1997, the Asian Financial Crisis and, second, the Global Financial Crisis between mid-2007 and early 2009 (Newell 2013; De Francesco and Hartigan 2009).

### 2.4. The Role of REITs and Their Relationship with Other Investment Vehicles

Based on the role of property, including generating long-term cash flow and the ability to capture inflation and enhance environmental, social and governance goals for institutional investors, REITs have been considered as alternative investment vehicles to common stock investment (Akinsomi 2020). The advantages of REITs are the diversification of portfolios, providing additional liquidity benefits, and tax benefits (EPRA 2021).

The role of portfolio diversification of REITs mainly relies on the risk-adjusted characteristic. Therefore, assessing the risk level has been thoroughly studied. De Francesco and Hartigan (2009) suggests that A-REITs have become more attractive in delivering a higher return but on the condition of bearing higher risk. The changing risk characteristics, such as rising gearing levels, increasing offshore exposure and evolving management structure to stapled trusts, have been factors. A-REITs have been depicted as a defensive style of investment (De Francesco and Hartigan 2009), and the volatility of REITs has been well-connected to the sensitivity of the stock market (Tsai et al. 2010; West and Worthington 2006; Danso 2022; Lee 2009; Block 2012).

As securitized investment vehicles, REITs' returns have been discussed for decades. Their association with other investment assets has also been examined through integrated econometric methods and approaches (West and Worthington 2006; Clayton 2003; Okunev et al. 2002; Geltner et al. 2014; Allen et al. 2000; Li 2018). One commonly evaluated and accepted perspective is that the performance of REITs shows a stronger correlation to the stock market in the short run but is more associated with the direct real estate market in the long term. The risk, sensitivity, volatility, or hedging effectiveness, whichever aspect of REITs has been assessed, the data from the return indices has been collected to be input into statistical models.

The initial public offering (IPO) performance of REITs has also been another topic of study (Dimovski et al. 2019). The forecasted dividends of A-REITs are considered to be

biased by Dimovski et al. (2019), which has been attributed to institutional factors during the IPO forecasting process. The absence of institutional support at the outset of an A-REITs' IPO is the reason. Mensi et al. explore the quantile return spillovers between oil and foreign REIT markets utilizing a quantile connectedness technique; their findings indicate that the extreme oil–REIT connection is diverse and asymmetric. At lower quantiles, the return spillover is greater. In addition, the oil market operates as a net transmitter of return spillovers to the REIT markets during periods of negative returns and as a net receiver during periods of positive returns. The hedging method was costly during COVID-19, with oil giving Hong Kong the greatest hedging effectiveness (Mensi et al. 2022).

Apart from the commonly accepted viewpoints, the emerging sector-specified A-REITs have been assessed and found to be more resilient compared with traditional sector REITs (Lin et al. 2019; Akinsomi 2020). However, these studies have been confined within the scope of the REITs themselves, along with discussions on the related REITs' performance.

*2.5. Mixed Outcome of Direct-Indirect Real Estate Relationship Studies*

The evaluation of the relationship between securitized real estate markets and other asset markets are mixed. Cheong et al. (2009) states that the Australian securitized real estate market shows no apparent connection with the stock market, in the long or short term. On the contrary, by investigating the dynamics of the connectiveness among the direct real estate market, securitized market, and the stock market by applying the connected index approach, Nguyen et al. (2022) noted that the securitized real estate market is more linked with its underlying asset market. How was this connection built and developed has yet to be discussed. The fundamental assets are essentially the same in the listed property market and general stock market (Hoesli and Oikarinen 2012).

## 3. Methodology

Even though REITs are traded in the public stock market, and therefore can be assessed as a certain type of security, there are two tests for companies to pass as forming REITs, which make them more bonded to the direct real estate market than the overall stock market. First, REITs are required to distribute at least 90% of their net income as dividends. Second, the bulk of assets of any individual REIT should be directedly related to real estate properties. A regular corporation makes a profit and pays taxes on its entire profit, and then decides how to allocate its after-tax profits between dividends and reinvestment. A REIT simply distributes all or almost all its profits and does not pay tax. In the case of A-REITs, the first test is applicable as, by doing so, A-REITs are exempt from corporate tax. For the second test, unlike the general rule in the U.S., where at least 75% of a REIT's assets should consist of real estate assets, an A-REIT must satisfy the conditions of being managed investment trusts (MIT) under the Corporations Act 2001 (Commonwealth 2001). A-REITs must and can only operate based on the following definition:

> [ . . . ] 'eligible investment business' covers investing in 'land' for the purpose of deriving rent and/or investing or trading in various financial instruments including units in unit trusts, shares in companies' loans and derivatives.[1] (EPRA 2021)

So, the valuation of A-REITs' performance is inevitably connected to the valuation of their underlying property assets. Three models are well applied to evaluate REIT performance and direct property performance. The first is the Net Asset Value (NAV) method[2]. The second one is the comparable sales method. The third one is based on discounting the future operating cash flows of the REITs, as well as of the underlying properties, to the present value (the DCF method).

These three methods are normally consolidated and applied during the appraisals.

This study uses a three-step method based on the key concepts extracted from the three models introduced above. The first two steps of this method are the theoretical models for valuation. Step one is the REITs; earnings measurement summarized by Geltner et al. (2014) (Table 1). REITs earnings measures are a basic but widely used approach to assess

the stream of cash flow produced by a REIT. The main consideration of REITs earnings measures is that, compared with direct property cash flow generation, the application of AFFO (adjusted funds from operations) differentiates the earnings of REITs from the EBTCF (equity-before-taxes cash-flow) of direct property earnings. We are able to use the process of transferring direct property earnings into REITs earnings under the rule that 90% of REITs net income should be distributed as dividends. This step is the stage where we introduce the income from direct property to the REITs valuation process.

**Table 1.** Example of REITs Earning Measures.

| Direct Property | | REIT | |
|---|---|---|---|
| PGI | 180 | PGI | 180 |
| Vacancy | −9 | Vacancy | −9 |
| Operating Expenses | −71 | Operating Expenses | −71 |
| | | NOI (property level) | 100 |
| NOI | 100 | General and Administrative Expenses | −3 |
| | | EBITDA | 97 |
| Interest Expense | −40 | Interest Expense | −40 |
| | | FFO | 57 |
| Depreciation Expense | −20 | Depreciation Expense | −20 |
| GAAP Net Income | 40 | GAAP Net Income (Dividends ≥ 0.9 × Net Income = 33.3) | 37 |
| Add back Depreciation | +20 | Add back Depreciation | +20 |
| CapEx | −15 | CapEx | −15 |
| EBTCF | 45 | AFFO (FAD) (Available for plowback up to: 8.7) Div/FFO as low as 33.3 ÷ 57 = 58%, Div/FAD = 33.3 ÷ 42 = 79% | 42 |

The second step is based on the discounted cash flow method. This study will use a more specified model known as the Gordon growth model (the GGM). The GGM looks at a REIT as a stable and infinite stream of cash flow with a constant growth rate in the annual dividend. The following formula is the definition of the GGM:

$$E_0 = \sum_{t=1}^{\infty} \frac{DIV_t}{(1+r)^t}$$

where $E_0$ stands for the current value of the REIT's equity; $DIV_t$ represents the annual dividend expected to be distributed by year $t$; and $r$ is the market's required long-run total return expectation for investments in the REIT's shares.

Table 2 explains three main possible cases of REITs' growth (Geltner et al. 2014). In these cases, adjusted funds from operations (AFFO), instead of dividends, are applied in the Gordon growth model. The major consideration of this arrangement is that the REITs' earnings are able to be directly applied. Theoretically, the REITs' earnings are at the same level as the same-store earnings from the direct property assets.

Steps one and step give us the theoretical valuation outcome of the REITs. However, in practice, there are other aspects that impact the values of a REIT. The theoretical outcome may not be completely consistent with the publicly reported outcomes (COF 2021, 2022a, 2022b). The disparities detected in steps one and two will be investigated in step three.

In the case of this study, the hypothesis is that, apart from the rental income of the underlying property assets, there are features of the underlying property assets indirectly linked to the value of the REIT. The examples of these features are:

(a)  Tenant composition (the business sectors of the tenants),
(b)  Tenant profile by occupied area,
(c)  Weighted average lease expiry (WALE),
(d)  Capitalization rate/weighted average capitalization rate,

(e)    Age of building,

(f)    Occupancy rate,

(g)    NABERS[3] rating.

**Table 2.** Three cases of GGM of REIT Share Prices.

| Assumptions about Growth Environment | Comments |
|---|---|
| Case 1: No Expansion [no plowback ($p = 0$), $DIV_1 = AFFO_1$] $$E = \frac{DIV_1}{r - g_E} = \frac{AFFO_1}{r - g_E}$$ | REIT as a passive, pass-through entity that owns a static portfolio of properties. DIV growth, $gE$ is growth in EBTCF from existing assets in place; growth in same-store levered property income. |
| $g = g_E + P(r - g_E) = g_E + pyE$ Case 2: Internally Financed Expansion but No Growth Opportunities $$E = \frac{DIV_1}{r - g} = \frac{(1-p)AFFO_1}{r - g} = \frac{AFFO_1}{\gamma - g_E}$$ $0 < P < 1$ | REIT grows by reinvesting $p\%$ of AFFO each year; DIV is less than AFFO. REIT buys properties identical to the ones it currently owns, at market value (i.e., NPV = 0), using retained AFFO and debt, keeping a constant debt/equity ratio → REIT equity value is unchanged from case 1. Div. growth, $g$, exceeds same store EBTCF (and AFFO) growth, $gE$ but the REIT's price/earnings (E/AFFO) ratio is same as in Case 1. |
| Case 3: Internally Financed Expansion and Growth Opportunities $$E^* = E + NPV(growth\ opportunities)$$ $$E^* = \frac{(1-p)AFFO_1}{r - g} + NPV(growth\ opportunities)$$ $$= \frac{AFFO_1}{\gamma - g_E} + NPV(growth\ opportunities)$$ $$E^* = \frac{DIV_1}{r - g^*} = \frac{(1-p)AFFO_1}{r - g^*}$$ | Here, REIT is perceived to have the ability to find and execute NPV > 0 deals or projects, possibly at times due to differential pricing in public versus private real estate markets. $g^*$ incorporates future increases in AFFO due to such positive NPV growth opportunities into the growth rate in the GGM; it "merges" the impact of growth opportunities into $gg$, thus $g^* > g$, and the Case 3 REIT's price/earnings (E\*/AFFO) ratio is greater than that of Cases 1 or 2. |

Step three will construct a linear regression analysis following the framework of the classical hedonic pricing model. Hedonic analysis is originally the study of the relationship between the price of a product and the characteristics of the product. It was first applied in 1926 to value the price of farmland (Olanrele et al. 2014). The hedonic pricing model has been widely discussed academically and accepted as a way of assessing the price of real estate which has a variety of attributes. Previous research constructed the hedonic pricing model, which can be directly applied to real estate practice (Ishijima and Maeda 2015; Sirmans et al. 2005). The model generally takes the form:

Price = $f$ (Physical characteristics, Other factors)

The definition of this function is that the price of a house is a function of its physical characteristics and other factors such as school quality (Sirmans et al. 2005). Dubreuille et al. (2016) then used this model to examine the U.S. REITs market by making the form a linear mathematic function.

$$y = \sum_{i=1}^{n} \beta_i x_i + \varepsilon_i$$

where $y$ represents the price of the REIT; $x_i$ stands for the independent variable features listed above; $\beta_i$ represents a variation in the price of the asset being a consequence of an increase of one unit of the value of one or more variables. The function requires a statistical procedure to calculate the values of $\beta_i$, which illustrates the influence of each $x$ on the price. The hedonic pricing model is useful to establish the relationship between share prices and net asset values for investment purpose (Roubi and Litteljohn 2004).

*Hypothesis*

Based on a preliminary evaluation of the performance indicators of COF (the Fund) and its underlying assets (Table 3), this study determined that, since the inception of COF, both the Fund and its underlying assets have grown, albeit at varying rates (MorningStar

2002). We indicate that, between the direct income-generating process of the underlying assets and the dividend production of the Fund, operational expenditure, debt obligations, gearing, administrative and management fees have all influenced the observed outcome. On the side of the underlying assets, changes in the portfolio's net lettable area, rent rate, occupancy rate, weighted average lease expiration date, and even tenant structure may have had a role. This association will be examined in-depth in the Analysis section of this study.

**Table 3.** Preliminary COF Performance Comparison.

| Item | 06/15 | 06/16 | 06/17 | 06/18 | 06/19 | 06/20 | 06/21 |
|---|---|---|---|---|---|---|---|
| Portfolio Property Rental Income | 18,757,117 | 39,536,000 | 41,385,000 | 777,025,000 | 108,859,000 | 146,341,000 | 161,805,000 |
| | | 111% | 5% | 86% | 41% | 34% | 11% |
| Non-Current Asset Value-Property | 323,110,000 | 398,730,000 | 609,950,000 | 836,300,000 | 1,321,475,000 | 2,085,650,000 | 2,046,221,000 |
| | | 23% | 53% | 37% | 58% | 58% | −2% |
| COF Market Capitalization ($m) | 241.87 | 251.77 | 440.81 | 596.76 | 988.74 | 1034.45 | 1198.32 |
| | | 4% | 75% | 35% | 66% | 5% | 16% |

## 4. Data and Results

### 4.1. The Selection of Subject A-REIT

This research focuses on the Centuria Office A-REIT (ASX Code: COF). COF is the largest A-REIT office unit (by market cap on 31 December 2021). COF is a pure office REIT with 23 office properties spread in the six metropolitan cities of Australia. As the cash flow production of office property assets is uncomplicated, office-sector-specific REITs are a very steady security type. From the standpoint of direct property assets, the office properties in COF's portfolio are high-quality office properties that generate rental income effectively. The central business districts of major cities are not a limiting criterion for COF property selection. In addition, its portfolio includes both subregional centers and city-periphery business parks. This provides the COF with a portfolio structure that is

The COF was first officially listed in the Australian Stock Exchange on 10 December 2014, recorded as Centuria Metropolitan REIT (Security code: CMA). It owned a combination of both office and industrial properties, with 90% (by area) of its properties designated as offices. Since 2019, CMA has specialized in office properties. On 10 February 2020, CMA changed its name to COF, as it solely focuses on office property assets.

### 4.2. Data Illustration

The data (Table 4) is extracted from the COF Annual Report from the financial years 2015 (FY2015) to 2021 (FY2021).

For step one, the data from REITs earning measures are collected from the Financial Statement sections of the COF Annual Reports. Omitting minor changes in the recorded items every year, there are 71 total items in the Financial Statement annually.



**Table 4.** Variables Summary.

| Financial Year | Total Property Asset Book Value ($m) | Portfolio Cap Rate | Net Lettable Area (sqm) | Property Occu-pancy (By lncoms) | Property WALE | Weighted Average Building Age (By value) | Weighted Average NAERS Energy Rating (Stars) | Weighted Average NAERS Water Rating (Stars) | COF Gearing | COF FFO ($m) | New Leases (sqm) | Renewal Leases (sqm) | ASX-Listed Tenant Composi-tion (By Rental lncome) | Government Tenant Composition (By Rental lncome) | Dividend (Cent per Share) |
|---|---|---|---|---|---|---|---|---|---|---|---|---|---|---|---|
| | | | Independent Variables | | | | | | | | | | | | Dependent Variable |
| 2022 (HY) | 2254 | 5.65% | 302,700 | 94.30% | 4.3 | 16.87 | 4.8 | 3 | 33.10% | 54.7 | 4854 | 13,816 | 27% | 25% | 4.15 |
| 2021 | 2014 | 5.81% | 287,007 | 93.10% | 4.3 | 17.27 | 4.8 | 3.3 | 33.50% | 233 | 26,388 | 25,689 | 26% | 27% | 4.12 |
| 2020 | 2053 | 6.93% | 304,586 | 98.10% | 4.7 | 16.32 | 4.8 | 3.3 | 34.50% | 85.4 | 5392 | 26,987 | 25.10% | 25.40% | 4.45 |
| 2019 | 1400 | 6.22% | 218,080 | 98.40% | 3.9 | 15.8 | 4.6 | 3.2 | 34.20% | 61.2 | 5463 | 16,295 | 27% | 11% | 4.36 |
| 2018 | 930.5 | 6.68% | 183,339 | 98.90% | 4 | 19.51 | 4.0 | 2.2 | 28.30% | 44.1 | 6985 | 10,985 | 30% | 15% | 4.53 |

These 71 items are summarized into five categories:

1. Revenue, including rental income (accounted for over 80% of the total revenue), gain on derivative financial instruments, and interest income.
2. Expenses, including rates, taxes, and other property outgoings, management fees, financial costs, loss on fair value of investment properties, and other expenses. The largest proportion of the Expenses category normally goes to statutory expenditures such as rates and taxes, administrative expenditures such as management fees, and financial costs.
3. Assets, including current assets and non-current assets. Current assets mostly consist of cash (and cash equivalents) and investment properties held for sale. The major part of non-current assets is investment properties, which accounts for almost 100% of the total.
4. Liabilities, including current and non-current liabilities. Current liabilities are trade and distribution payables. Non-current liabilities are mostly borrowings.
5. Equity. Equity consists of issued capital and retained earnings (or accumulated losses).
6. Cash flows. Cash flows are recorded in three streams: cash flows from operating activities, from investing activities, and from financing activities. Net cash generated by operating activities and proceeds from borrowings are the major cash inflows. Repayment on borrowings is the major cash outflow.

The data related to the direct property in COF's portfolio is extracted from the COF Annual Report (COF 2021) and the COF Property Compendium (COF 2022b). As there exist constant small changes in the Property Directory of the COF every year, this study will not investigate the direct property asset data property-by-property. The yearly overviews of the property portfolio are summarized, and the features that affect the valuation of the REIT are organized.

The key metrics of direct property assets include the number of assets, portfolio book value, capitalization rate, net lettable area, occupancy rate, WALE, and average building age. The tenant composition, property geographic location, and tenant profile are other features to be investigated.

For stage three, the multiple regression analysis with the hedonic pricing model framework, the dependent variable is the price of the COF REIT. The independent variables are divided into two sets: set one—REIT level independent variables, and, set two—direct property asset-level independent variables.

Specifically, the independent variables at the REIT level are:

- Funds from operations (calculated by step one, REITs earning measures),
- Dividend per unit (recorded in the Financial Statements),
- Average yield on dividend per unit,
- Gearing (defined as total borrowings less cash divided by total assets less cash and goodwill, recorded in the Financial Statements),
- Yearly total security holder return.

At the direct property asset level, the independent variables are:

- Property asset book value,
- Property occupancy rate,
- Portfolio WALE,
- Average building age,
- Sustainability ratings.

### 4.3. Result

To establish the regression analysis, we first investigate if there is a linear regression relationship between each independent variable and the dependent variable. Considering the limits of this study, the regression analysis is conducted in a straightforward manner, in which the relationships between the independent variables and the dependent variable is examined one by one (Figures S1–S14, in the Supplementary Materials).

The preliminary scatter diagrams are the tools to help with the illustration. The straight trend line of each scatter diagram is used for observing the degree of concentration. Where the scatters are located closer to the trend line, the more significant linear relationship is exhibited.

The regression statistics summarize the quantitative extent of the relationship between each variable (feature) and the dividend. In the regression statistics, ***Multiple R*** stands for the regression coefficient, of which the value is between −1 and 1. The closer it is to −1, the higher the negative correlation, and vice versa.

***R Squared*** stands for the fit coefficient. It is the square of R, and the value is between 0 and 1. The larger the value is, the higher the fitting degree of regression model and actual data is.

***Standard error*** represents the distance between the actual value and the regression line. The smaller the value, the more accurate the regression model can be.

***Intercept Coefficients*** and ***X Variable 1 Coefficients*** are used for expressing the mathematical relationship between the variables and the dividend. In the case of this study,

the Dividend = Value of each variable × X Variable 1 Coefficients + Intercept Coefficient

At the A-REIT level, the portfolio book value, the overall portfolio capitalization rate, and the COF's gearing rate show a significant linear relationship.

## 5. Discussion

### 5.1. Results Discussion

After observing the first linear relationship, the fundamental quantitative linear regression analysis is applied. It is also undertaken on an individual basis. As the portfolio WALE does not appear to have a linear relationship with the payout, this aspect will be omitted from the analysis.

Among the A-REIT level variables (Table 5), the portfolio capitalization rate is the most significant factor that affects the dividend, followed by the total property asset market value. Considering this, the valuation outcomes of the property assets make a relatively significant difference in the A-REIT's dividend. Despite the significance of the gearing in the capital structure of the REIT, it has the least impact on the dividend in this group.

**Table 5.** Regression results of A-REIT level features.

|  | Total Property Asset Book Value ($m) | Portfolio Cap Rate | COF Gearing | COF FFO ($m) |
|---|---|---|---|---|
| Multiple R | 0.715 | 0.814 | 0.467 | 0.600 |
| R Squared | 0.511 | 0.663 | 0.218 | 0.360 |
| Adjusted R Squared | 0.349 | 0.551 | −0.042 | 0.147 |
| Standard Error | 0.146 | 0.122 | 0.185 | 0.167 |
| Intercept Coefficients | 4.730 | 2.116 | 5.417 | 4.455 |
| X Variable 1 Coefficients | 0.000 | 36.415 | −3.346 | −0.001 |

Among the direct property asset features (Table 6), the occupancy rate is highly correlated to the dividend, followed by the NLA. The weighted average building age is relatively insignificant in affecting the dividend.

**Table 6.** Regression results of direct property asset features.

| | Property Occupancy (By Income) | Net Lettable Area (sqm) | Weighted Average Building Age (By Value) |
|---|---|---|---|
| Multiple R | 0.954 | 0.616 | 0.347 |
| R Squared | 0.911 | 0.379 | 0.12 |
| Adjusted R Squared | 0.881 | 0.172 | −0.173 |
| Standard Error | 0.062 | 0.165 | 0.196 |
| Intercept Coefficients | −1.958 | 4.846 | 3.569 |
| X Variable 1 Coefficients | 6.504 | 0.000 | 0.044 |

The environmental rating features also less significantly affect the dividend of the COF (Table 7). However, the uplifting of the level of corporate social responsibility is observed to be helpful with the risk-adjusted performance of A-REITs.

**Table 7.** Regression results of environmental rating features.

| | Weighted Average NAERS Energy Rating (Stars) | Weighted Average NAERS Water Rating (Stars) |
|---|---|---|
| Multiple R | 0.692 | 0.582 |
| R Squared | 0.479 | 0.339 |
| Adjusted R Squared | 0.306 | 0.118 |
| Standard Error | 0.151 | 0.17 |
| Intercept Coefficients | 5.978 | 5.023 |
| X Variable 1 Coefficients | −0.361 | −0.232 |

Regarding the tenant composition aspects, normally ASX-Listed tenants and government tenants are considered as good quality, as they are reliable in maintaining stable long-term tenancies (Table 8). However, the regression analysis outcome suggests that, in determining the dividend, the tenant composition's effect on the dividend is unlikely a direct linear relationship.

**Table 8.** Regression results of tenant composition.

| | ASX-Listed Tenant Composition (By Rental Income) | Government Tenant Composition (By Rental Income) |
|---|---|---|
| Multiple R | 0.436 | 0.569 |
| R Squared | 0.190 | 0.323 |
| Adjusted R Squared | −0.080 | 0.098 |
| Standard Error | 0.188 | 0.172 |
| Intercept Coefficients | 3.165 | 4.618 |
| X Variable 1 Coefficients | 4.283 | −1.433 |

*5.2. Limitations*

There are two major limitations of this study. First, this study looks into only one specific A-REIT. The Centuria Office REIT is the subject A-REIT of this study. Compared with the overall ASX-listed real estate investment trust market, the COF only represents 0.79% of the total market capitalization in 2021 (ASX 2021). It is also a very recent A-REIT, founded in 2015.

The overall A-REITs market covers both the traditional property sectors, including retail, office, and industrial, and the innovative property sectors such as hotels and resorts, residential, and healthcare. Compared with the abovementioned A-REITs sectors, the COF is an office sector-specified A-REIT. The income pattern of COF is onefold, namely the rental revenue of the office buildings in its portfolio. The features of the direct property assets, which may affect the REIT's return, are also relatively straightforward. If further

studies intend to investigate the relationship between A-REITs' performance and the features of their underlying property assets, more real estate subsectors should be taken into consideration, as well as their own corresponding property asset features. Taking a retail-specified A-REIT as an example, the visitor numbers to shopping centers, retail turnover, and demographics of the total trade area are the features that have the potential to affect the study outcome.

The second limitation of this study lies in the simplified regression model that has been applied. This study only observed and assessed single linear relationships. The features selected as the variables could be improved. If the scope of the A-REITs studied is to be extended with a more representative number of A-REITs, the analysis model needs to be more rigorous as the hedonic analysis technique is driven by data, not theory (Isakson 2001). To complete a proper multiple regression analysis consists of three stages: 1. analyzing the correlation of the data, 2. estimating the model, and 3. evaluating the validity and usefulness of the model.

Meanwhile, the regression analysis method has its own pitfalls. Isakson (2001) explains the two disadvantages,

- The model specification,
- The robustness of the results of the regression,

The model specification primarily is concerned with the choice of dependent and independent variables, the formula form of the correlation and the statistical significance of the independent variables. The robustness of the result of the regression analysis is concerned with the measurement errors in the data and sensitivity of the results to changes.

To avoid the pitfalls discussed above, a better option is to use more data of each independent variable in the multiple regression analysis model (MRA). If the data of the independent variables are fewer than ten items, they can lead to a misrepresentation of the results, as MRA is a data-hungry statistical technique (Isakson 2001).

## 6. Conclusions

The discussions have suggested that there exists a certain degree of linear correlation between the underlying property assets and the return of the subject A-REITs. The most significant variable is the occupancy of the offices. The higher the occupancy rate, the better the dividend can be. The relatively fixed, physical features of the properties are less likely to directly affect the dividend. These features include the net lettable areas and sustainability ratings. The newer the property, the higher the dividend can be. This is not just because of the building's age but also about the frequency of the renewal of the building. The features at the A-REIT level also affect the dividend outcomes, specifically the total portfolio market value and the capitalization rate. This suggests that the annual valuation outcomes show a positive relation with the performance of the A-REIT. Further study could be conducted to discuss how these relationships are built.

Lease management is one factor that affects office property occupancy. One way to avoid the risk of occupancy uncertainty is preleasing (Buttimer and Ott 2007). For individual buildings, the acceptable vacancy rates are higher in weaker economies, because of the slower arrivals of tenants to fill the vacant space. For the market vacancy rate, strong economies generally have higher vacancy rates, due to an increasing number of speculative buildings.

The connection between the building age of the direct property assets and the COF return may be reflected in the depreciation of the property value. Normally, the depreciation of assets is discussed from the perspective of tax matters and investment management. There are three sources of decline in the real value of property that contribute to depreciation: the physical, functional, and economic obsolescence of the building structure (Bokhari and Geltner 2018).

Jang et al. found that REITs market capitalization has been significantly affected by gross domestic product per capita, the overall stock market capitalization, and the stability of the banking sector at the country level (Jang et al. 2020). Experiences from the U.S. real

estate market have shown that traditional variables, such as return expectation, and risk premiums, and non-traditional variables, such as the degree of unemployment and past capitalization rate, are determinants of the capitalization rate (Larriva and Linneman 2022).

**Supplementary Materials:** The following supporting information can be downloaded at: https://www.mdpi.com/xxx/s1.

**Author Contributions:** Methodology, X.Z.; Investigation, X.L.; Resources, Y.Z.; Data curation, R.G.; Writing—original draft, X.L.; Writing—review & editing, X.Z.; Supervision, X.Z.; Project administration, X.Z. All authors have read and agreed to the published version of the manuscript.

**Funding:** This research received no external funding.

**Institutional Review Board Statement:** Not applicable.

**Informed Consent Statement:** Informed consent was obtained from all the subjects involved in the study.

**Data Availability Statement:** Data supporting reported results can be found in Supplementary Materials.

**Conflicts of Interest:** The authors declare no conflict of interest. The funders had no role in the design of the study; in the collection, analyses, or interpretation of data; in the writing of the manuscript; or in the decision to publish the results.

## Notes

[1] The definition of 'land' includes fixtures on the land and certain moveable property (e.g., chattels) customarily supplied, being a property that is incidental and relevant to the renting of the land and ancillary to the ownership and utilization of the land. Ineligible activities are regarded as trading activities.

[2] The net asset value (NAV) is an adjustment of the value of real estate assets based on the fair value of assets in the balance sheet. The value of the shareholders' equity is calculated by subtracting revalued liabilities from the fair value of assets. The following formula is used to define NAV: $NAV = \frac{Market\ Value\ of\ properties + Other\ Assets - Total\ Liabilities}{Number\ of\ Shares}$.

[3] NABERS stands for the National Australian Built Environment Rating System. It can be used to measure a building's energy efficiency and carbon emissions, as well as the water consumed and the waste produced, and compare it to similar buildings (NABERS n.d.).

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
