# Peer review of "Analyzing the Relationship between the Features of Direct Real Estate Assets and Their Corresponding Australian—REITs"

_ijfs, doi:10.3390/ijfs11010029_

Round 1

Reviewer 1 Report

The study explores in the present context a very interesting topic. The real estate market and the real estate assets have always been a good indicators of any crisis, which is nowadays a topic of particular interest. However, it must be stressed that the study itself focuses on a narrow area, which is likely to reach a small proportion of readers.
The study could be improved on a number of points. I suggest that the author should keep the background part of the introduction, and it is also worth keeping the aim of the study here (chapters 1.1 and 1.2). I then suggest that he should continue with the literature review. Although the author draws on a large body of literature, I nevertheless feel that the grounding in the literature is very weak. Here I suggest the author to explain in much more detail the literature he has worked with. After all, he has reviewed proportionally much more literature than the scope of the literature review itself. I would then suggest that the research questions should definitely be moved to the methodological part of the study. I would also move questions 1.4 and e1.5 of the introduction to the literature review. I would also move the hypotheses to the end of the methodological section. Unfortunately, in this form, the thesis itself is for me unintelligible and ad hoc. I would also suggest that you pay attention to the format of the article. When evaluating the data, I feel that the size of the figures used is very large, and I also feel that there is a lot of numericality. However, it is also noticeable that there is very little explanatory section following the figures to make them meaningful to the average reader. What I also miss is the conclusion section. Although Chapter 6 contains a dialogue section, this again seems to me to be just another way of the author explaining the results. It is very good that it shows the limitations of the research, but I do not see any conclusions that could have been drawn from this work.
I would definitely recommend that the author revises the paper in a more substantial way in order to make the paper itself more readable and interesting.

Author Response

Thank you for your comments on our manuscript entitled " Analyzing the relationship between the features of the direct real estate assets and their corresponding Australian – REITs" (ID: ijfs-2034377). Those comments are very helpful for revising and improving our paper, as well as the important guiding significance to our future research. We have studied the comments carefully and made corrections which we hope meet with approval. In the revised manuscript, the supplementary parts were all highlighted within the document by using red-colored text. The main changes are summarized as follows:

  1. The introduction section has been adjusted according to your suggestions. The original sections 1.1, 1.2, 1.3, have been retained, 1.4 has been reorganized into the literature study, and 1.5 has been reorganized into the last part of the methodology.
  2. Added a seventh section at the end of the article: Conclusion.
  3. For the article to read smoothly, we consulted a language expert and we modified the language expressions in the article.

Once again, thank you very much for your constructive comments and suggestions which would help us in depth to improve the quality of the paper.

Reviewer 2 Report

Overview

The author investigates the determinants of Australian Real Estate Investment Trust. Using linear regression analysis, the author reports that the rate of office occupancy is the most determinant factor of REIT returns. In addition, the author finds that capitalization rate and property asset management book value are the main drivers of dividends, in a portfolio context.  

General comment

The analysis has not been performed in a concise and relevant manner. There is a lack of a good flow of information in all sections of the manuscript, which makes it recommendable under major conditions. The concerns and reservations about several aspects of the paper are detailed below.

Specific comments:

1.      The author should avoid using subsections for the Introduction and Literature review sections

2.      The author needs to present a solid motivation for the study

3.      The author should present the literature review in more details. For instance, and since the author claimed the existence of connectedness between the REIT market and the direct property investment market, the literature review should thoroughly discuss early findings in a complete and concise manner. Besides, there must be other types of connectedness between REIT and other markets, and the author should briefly go over such connectedness. For instance, see Mensi et al. (2022) Quantile connectedness and spillovers analysis between oil and international REIT markets. Finance Research Letters, 48, 102895.

4.      The methodology section is somehow confusing. The author stated that three methods are being used in his analysis, namely NAV, GGM, and linear regression analysis. If the author is convinced that the Hedonic Pricing Model is suitable for his analysis then he must empirically show the shortcomings of NAV and GGM.

5.      The results should be better presented. It is difficult to read Table 4. Figure 1 to 14 should be grouped together (in smaller sizes) into 2 or 3 separate figures.

6.      The discussion of results should be compared with early findings for corroboration.

7.      The discussion of results should be followed by implications.

8.      Add a conclusion section and include the limitations within.

9.      Although there are no typos, the manuscript should be language-edited for clarity and careful proofreading.       

Author Response

Thank you for your comments on our manuscript entitled " Analyzing the relationship between the features of the direct real estate assets and their corresponding Australian – REITs" (ID: ijfs-2034377). Those comments are very helpful for revising and improving our paper, as well as the important guiding significance to our future research. We have studied the comments carefully and made corrections which we hope meet with approval. In the revised manuscript, the supplementary parts were all highlighted within the document by using red-colored text. The main changes are summarized as follows:

  1. We have added studies from the literature review based on your suggestions, and specifically added Mensi et al. (2022) research findings to the discussion.
  2. The introduction section has been adjusted according to your suggestions
  3. We emphasized the motivation for the study in our research objectives.
  4. We have modified Table 4 to make it better to read
  5. Added a seventh section at the end of the article: Conclusion.
  6. For the article to read smoothly, we consulted a language expert and we modified the language expressions in the article.

Once again, thank you very much for your constructive comments and suggestions which would help us in depth to improve the quality of the paper.

Round 2

Reviewer 1 Report

The paper is much improved from the original version, but I still find the amount of figures too much and the size of the figure with the accompanying explanation not appropriate. Perhaps the authors should consider placing the figures in an appendix and referring to them in the text, so that the disproportion between the two parts is more evenly distributed.
The current data section should be renamed data and results.

Author Response

Thank you again for your comments on our manuscript entitled " Analyzing the relationship between the features of the direct real estate assets and their corresponding Australian – REITs" (ID: ijfs-2034377). Those comments are very helpful for revising and improving our paper. We have studied the comments carefully and made corrections which we hope meet with approval. The main changes are summarized as follows:

  1. Renamed the current data section : 4. Data and Results
  2. Placed Figures 1-14 in 4.3 Results in the Appendix-Figures and referred to them in the text.

Once again, thank you very much for your constructive comments and suggestions which would help us in depth to improve the quality of the paper.

Reviewer 2 Report

I am content with the revision made

Author Response

Thank you for your approval of the article revision.